# Nitric Oxide Nano-Delivery Systems for Cancer Therapeutics: Advances and Challenges

**DOI:** 10.3390/antiox9090791

**Published:** 2020-08-26

**Authors:** Long Binh Vong, Yukio Nagasaki

**Affiliations:** 1School of Biomedical Engineering, International University, Ho Chi Minh 700000, Vietnam; 2Vietnam National University Ho Chi Minh City (VNU-HCM), Ho Chi Minh 700000, Vietnam; 3Department of Materials Science, Graduate School of Pure and Applied Sciences, University of Tsukuba, Tsukuba, Ibaraki 305-8573, Japan; 4Master’s School of Medical Sciences, Graduate School of Comprehensive Human Sciences, University of Tsukuba, Tsukuba, Ibaraki 305-8577, Japan; 5Center for Research in Isotopes and Environmental Dynamics (CRiED), University of Tsukuba, Tennoudai 1-1-1, Tsukuba, Ibaraki 305-8573, Japan

**Keywords:** nitric oxide, drug resistance, self-assembled nanomedicine, gaseous signaling molecule, angiogenesis, cancer

## Abstract

Nitric oxide (NO) plays important roles in various physiological and pathological functions and processes in the human body. Therapeutic application of NO molecules has been investigated in various diseases, including cardiovascular disease, cancer, and infections. However, the extremely short half-life of NO, which limits its clinical use considerably, along with non-specific distribution, has resulted in a low therapeutic index and undesired adverse effects. To overcome the drawbacks of using this gaseous signaling molecule, researchers in the last several decades have focused on innovative medical technologies, specifically nanoparticle-based drug delivery systems (DDSs), because these systems alter the biodistribution of the therapeutic agent through controlled release at the target tissues, resulting in a significant therapeutic drug effect. Thus, the application of nano-systems for NO delivery in the field of biomedicine, particularly in the development of new drugs for cancer treatment, has been increasing worldwide. In this review, we discuss NO delivery nanoparticle systems, with the aim of improving drug delivery development for conventional chemotherapies and controlling multidrug resistance in cancer treatments.

## 1. Biological Functions of Nitric Oxide and Therapeutics in Cancer

Nitric oxide (NO), an important cellular signaling gaseous molecule that diffuses easily in a biological environment, plays multifunctional roles in physiological and pathological processes. In the healthy body, NO is produced via the enzymatic reaction of endogenous L-arginine (L-Arg) by endothelial nitric oxide synthase (eNOS), which is highly expressed in the endothelium. One of the other isoforms of NOS is neuronal nitric oxide synthase (nNOS), which is mainly expressed in neurons in brain tissue. In the vasculature, NO molecules modulate vasodilation and blood flow by regulating the soluble guanylate cyclase in the vascular smooth muscle cells [1,2]. The abnormal physiological generation of NO leads to the dysfunction of the endothelium, resulting in various cardiovascular diseases, such as hypertension, atherosclerosis, and heart failure. In the brain, NO also controls blood flow and angiogenesis and plays a role in behavior and cognition functions as an important neurotransmitter [3,4]. In pathological conditions, such as inflammation and cancer, the activated macrophages elevate the expression of inducible nitric oxide synthase (iNOS) with a strong ability to produce NO and regulate immune responses [5,6]. The generated NO can also be used by immune cells as a toxic defense molecule to kill the infectious microorganisms [7,8]. The biofunctional action of NO in the immune system is to regulate the functional activity and proliferation of various immune cells [9]. Whereas eNOS and nNOS are constitutively expressed and exhibit their activity by binding with calmodulin in the presence of the intracellular Ca^2+^ ion, iNOS is inducible and its activity is independent of the level of Ca^2+^ in the cell because it binds to calmodulin regardless of intracellular Ca^2+^ ion concentration [10]. It is also reported that NO exhibits different functions in the biological features of a cancer cell [11,12]. Depending on the concentration, NO displays dichotomous roles—it can promote the development and progression of cancer or it may exhibit anti-cancer activity and inhibit cancer cell growth (Figure 1) [6,13]. At lower concentrations (100–500 nM), NO induces angiogenesis, which promotes cancer cell proliferation and tumor progression, invasion, and metastasis. NO stimulates several signaling pathways, such as the vascular endothelial growth factor (VEGF) and epidermal growth factor receptor (EGFR), which are involved in promoting neovascularization or angiogenesis [14,15]. In contrast, at higher concentrations (>500 nM), NO causes DNA damage, apoptosis, and cell death via the formation of toxic peroxynitrite [12,16]. In the physiological condition, NO basically undergoes a rapid oxidation to form nitrite anion (NO_2_^−^), which is further oxidized to nitrate anion (NO_3_^−^). The autooxidation of NO molecules also occurs to generate N_2_O_3_ via the formation of nitrite radical (NO_2_^•^), and this is a reversible reaction [17,18]. Because NO has various functions in the cancer microenvironment, versatile and unique cancer treatments using NO have been developed recently [13,19,20]. The biosynthesis of NO and its biological action are described in Figure 1. Although NO plays many important roles, its half-life is extremely short (ranging from 0.09 to 2 s [21]), which limits its therapeutic applications considerably. In the next section, we focus on the biofunctional action of NO in cancer therapeutics, and various NO delivery nano-systems are presented, with the goal of improving the efficiency of cancer treatment.

## 2. Nanoparticle Systems for Controlled Release of NO Molecule in Cancer

As described above, owing to its extremely short half-life and rapid diffusivity, the direct use of NO for therapeutic targets is challenging. Because a solid tumor is located in specific organs, it is important to spatiotemporally control the distribution and release of NO molecules in the body. Although many NO donors have been developed and investigated to fight cancer (Figure 2) [24,25], their therapeutic efficiencies are not always satisfactory owing to low biological stability and non-specific dispersion, resulting in low bioavailability of these low-molecular-weight (LMW) agents in vivo. To overcome the drawbacks of the LMW NO donors, researchers in the last several decades have paid considerable attention to nanoparticle-based drug delivery systems (DDSs) as a new medical technology, given that these systems can alter the biodistribution of therapeutic agents and control drug release, resulting in a significant therapeutic effect of NO. Versatile nano-DSSs, such as silica, metal oxide, and polymeric nanoparticles, as well as liposomes and dendrimers, have been used to deliver NO or NO donors to induce NO production for various therapeutic applications, such as wound healing, anti-bacterial therapy, and the treatment of cardiovascular disease and cancer (Figure 3) [26,27]. With the use of these nano-carrier systems, the release of NO molecules can be controlled spatiotemporally in vivo.

Because most of the NO donors immediately release NO under physiological conditions owing to their low biological stability (i.e., decomposition and uncontrolled release from the carriers), simple encapsulation of the conventional LMW NO donors does not result in adequate stability of the molecule and proper control of its release. Suchyta and Schoenfisch designed several NO donors from *N*-diazeniumdiolate homologs and extended the half-life (t_1/2_) of NO from several seconds to half an hour in aqueous media [28,29]. They succeeded in sustaining the release of NO by encapsulating the stable *N*-diazeniumdiolate in a liposomal nanoparticle in phosphate buffer saline (PBS) up to 48 h. The release kinetics of NO from liposomes was highly associated with the surface area of the lipid head groups and the hydrophobicity of the hydrocarbon chains in the phospholipid molecules of the designed liposomes. Although the stable NO donor was encapsulated in the liposome and was notably more stable than the conventional NO donor in PBS solution or serum medium, the NO donor itself was rapidly leaked from the liposome into the blood. Therefore, the stability in the blood was not always sufficient despite the sufficiently high stability in vitro, which might be due to the insufficient stability of the liposome itself in blood and/or the complex composition of the blood—specifically the very large number of proteins, lipids, and ions—which might also affect the stability of the particles [30]. Such factors must be taken into consideration when designing nano-systems for NO delivery.

To achieve the spatiotemporal control of NO release in vivo, a stable nanocarrier system and control in release-timing are important. Currently under development are several stimuli-responsive systems that trigger NO release in response to several stimulations, such as light and environmental pH. Kanayama et al. reported photo-triggered NO generation from a self-assembling polymer micelle with 4-nitro-3-trifluoro-methylphenyl units in its hydrophobic segment. Because the NO-releasing moiety is covalently conjugated to the amphiphilic block copolymer, it can avoid leakage of the NO molecule from the micelle without photo-irradiation [31]. Choi et al. used calcium phosphate to stably entrap the NO donor in a light-induced NO release system. The NO donor, diazeniumdilate, and a photo-sensitive proton donor, 2-nitrobenzaldehyde (ο-NBA), were mixed in calcium phosphate and encapsulated in the core of a mesoporous silica nanoparticle [32]. Under light irradiation, the proton generated from ο-NBA induced the degradation of calcium phosphate, followed by a release of the NO molecule in the target tissue. These are two examples of the successful stable encapsulation of a NO-precursor in a nanoparticle; however, these systems employed UV light as the stimulus for the cleavable reaction that releases NO or protons, which is not suitable in cases where light penetration to target tissues that lie deep under the skin is not possible. Such systems may be applicable only in cases where the tumor tissue is located on the surface of the organ, such as skin, gastric, and lung cancers. To improve the penetration of light to tissues that lie deep under the skin, avoiding the absorption by hemoglobin in the blood, a longer wavelength light source was investigated as a stimuli-responsive NO-releasing system. Several NO-releasing nano-systems that are responsive to near-infrared (NIF) light irradiation have been developed and have exhibited effective inhibition of tumor progression by the combined NO and photothermal therapy (PTT) [33,34,35]. Zhang et al. designed a bismuth sulfide (Bi_2_S_3_) nanoparticle loaded with the hydrophobic NO donor (*N,N′*-di-sec-butyl-*N,N′*-dinitroso-1,4-phenylenediamine (BNN)) via hydrophobic interaction. Because Bi_2_S_3_ is known to induce heat by the NIR irradiation of PTT, BNN subsequently releases the NO molecule, thus achieving the synergistic cancer therapy. An up-conversion nanoparticle, which emits UV-light from two weak NIR excitations, was also employed in a photo-induced NO delivery system [36].

Ultrasound is another technique used to develop stimuli-responsive NO-releasing systems [37,38]. Zhang et al. recently reported an ultrasound-triggered NO release system using a hollow mesoporous silica nanoparticle [39]. The authors explained that the ultrasound irradiation generates reactive oxygen species (ROS), which oxidize the encapsulated L-Arg monomer to generate NO to suppress the growth of a Panc-1 tumor. Although L-Arg is reported to generate NO by the oxidation reaction, its mechanism is not entirely clear [40]. Because the NO molecules rapidly react with ROS and generate other formulas, the detected NO might be generated by other mechanisms, such as an enzymatic reaction with iNOS, which is strongly expressed in tumorous macrophages. In addition, the hydrophilic L-Arg monomers might not be stably entrapped in the nanoparticles, although they are caught by the electrostatic interaction with anionic oligopeptide conjugated on the surface, which might cause the leakage of L-Arg during blood circulation. Since both NO and ROS present a double-faced role in the tumor environment, the generation of both NO and ROS is another approach to cause oxidative/nitrosative stress and cancer cell death [41,42]. The nanoparticles that simultaneously generate NO and ROS were reported to achieve the synergistic therapy to inhibit tumor progression by the formation of highly reactive nitrogen species [43,44,45,46]. Additionally, Sun et al. reported an antibody–NO donor conjugate via disulfide bonds to especially target the hepatocellular carcinoma, which highly expresses a cluster of differentiation (CD24) [47].

The development of stimuli-responsive NO-releasing systems in response to the tumor microenvironment has also been reported, viz., the release of NO can be triggered via the internal stimuli of the tumor tissue (such as glucose concentration, intracellular glutathione, enzyme, and acidic pH). Fan et al. developed a glucose-responsive system composed of a hollow mesoporous organosilica nanoparticle to co-deliver glucose oxidase (GOX) and L-Arg, which resulted in a remarkably improved anticancer effect [43]. When this nano-system is delivered to the tumor tissue, the GOX converts the glucose in the tumor into gluconic acid and H_2_O_2_, which further catalyzes L-Arg into NO molecules. The generated NO (approx. 5 µmol/L) was more than ten-fold higher under high-glucose concentration (1000 µg/L) than that under glucose-free media. These H_2_O_2_-NO-releasing nanoplatforms could minimize side effects, given that the release response only occurs under high-glucose and H_2_O_2_ environments, such as tumor tissue [48,49]. As mentioned above, however, the oxidation reaction of L-Arg to chemically form NO by H_2_O_2_ is not clearly confirmed, and a high concentration of reactants and catalysts is required to achieve rapid and efficient NO generation [50,51]. Besides, increased uptake of glucose by the Warburg effect enhances the intracellular glucose concentration in cancer cells, but not surrounding tumor microenvironment [52,53]; thus, it should be considered to deliver sufficiently both L-Arg substrate and GOX enzyme into the cancer cells. However, the other mechanisms might not be considered that the delivered L-Arg can be catalyzed into NO in the tumor microenvironment by the overexpressed iNOS from tumor-associated macrophage. Calcium carbonate is often utilized as a matrix of drug and/or genes because of the increased solubility in the acidic environment [54]. Lee et al. designed calcium carbonate-based nanocarriers to intracellularly release NO from S-nitroglutathione (GSNO, an NO donor). Under neutral blood conditions, the crystalline calcium carbonate nanoparticle stably entrapped the GSNO, whereas it erodes in the acidic endosome after internalization into cancer cells followed by the release of NO intracellularly, which significantly improves the anticancer activity of co-delivered doxorubicin [55,56].

One of the major problems with these nanocarrier systems is that physically entrapped NO donors and/or NO substrates, such as L-Arg often leak out before they reach their targets [57]. Recently, we developed a self-assembling polymer substrate system; in other words, by covalently introducing a substrate into the nano-system, the unwanted leakage of substrates during blood circulation is prevented. In this design, a poly(ethylene glycol)-*b*-poly(amino acid) block copolymer contains an L-Arg repeating unit in the main chain of the poly(amino acid) segment. The PEG-*b*-poly(L-Arg) block copolymer forms polyion complex (PIC) micelles composed of a positively recharged poly(L-Arg) segment and negatively recharged chondroitin sulfate for anticancer therapeutics (Figure 4A) [58,59]. The poly(L-Arg)-based PIC (abbreviated as Nano^ARG^) is approximately 30 nm in diameter, with a narrow size distribution. After intravenous administration, Nano^ARG^ can gradually accumulate at the tumor tissue via the enhanced permeability and retention (EPR) effect [60,61]. In the tumor microenvironment, the poly(L-Arg)-based PIC is taken up by the activated macrophage, followed by the liberation of L-Arg via hydrolysis with the highly expressed intracellular proteases. Because the tumor-associated macrophages highly express iNOS, the liberated L-Arg monomers effectively convert into NO (Figure 4A). Interestingly, the intravenous administration of Nano^ARG^ clearly shows the dose-dependent tumor progression and suppression in the tumor-bearing mice (Figure 4B). Namely, the progression of tumor growth is significantly increased when the Nano^ARG^ is administered once or twice daily by intravenous injection to cancer-bearing mice. However, three times administration over three consecutive days of Nano^ARG^ administration results in the same tumor growth rate as in the control, and four times administration (four consecutive days) decreases the progression of the tumor growth, indicating a very clear dose-dependence (Figure 4B); that is, the low doses increased the angiogenesis of the tumor tissues, whereas the high doses led to tumor-cell death by apoptosis. This consecutive double enzyme-responsive NO-releasing system, with a polypeptide self-assembling design, prevents the leaking of active components from nanocarrier during blood circulation. Currently, we are focusing on the construction of the concept “enzyme responsive self-assembled macromolecular drugs”, viz., the active components are incorporated as a component of the self-assembling polymer via covalent linkage to prevent the leakage of the drugs during blood circulation and to vary the biodistribution of the drugs, while it collapses at the target site due to highly expressed enzymes at the site [62,63,64]. Hubbell et al. also reported self-assembling block copolymer micelles containing an NO donor covalently conjugated with the hydrophobic segment; the NO-releasing rate based on the hydrolysis of *N*-diazeniumdiolate group significantly increased after the disintegration of the micelle [65].

## 3. NO-Delivery Nanocarriers Overcome Multidrug Resistance in Cancer

Multidrug resistance (MDR) of cancer cells remains a major challenge in cancer chemotherapy. Currently, the most accepted mechanism of MDR is decreased drug uptake and increased drug efflux by overexpression of ATP-binding cassette transporters including P-glycoprotein (P-gp) [66]. It is reported that NO decreases the ATPase activity of P-gp, resulting in suppression of MDR in cancer cell lines in vitro [67,68]. Fang et al. and Chung et al. independently reported that UV- and pH-responsive NO donors remarkably increased the uptake of the anticancer drug-encapsulated nanoparticles in an MDR ovarian cancer cell line by inhibiting the P-gp activity in MDR cancer cells [69,70]. A decrease of 50.1% of P-gp expression in the MDR breast cancer cell (MCF-7/ADR) was observed after treatment with a 10-hydroxycamptothecin-loaded chitosan-based nanocarrier having a glutathione-responsive NO release moiety [71]. NO has also been reported to influence the hypoxia-related MDR system in the tumor microenvironment. Due to the inadequate and poorly formed vasculature, tumor tissue forms hypoxia regions, which are highly associated with the development of chemoresistance via the activation of the hypoxia-inducible factor-1 (HIF-1), one of the transcription factors, promoting MDR expression under low oxygen conditions [72,73]. NO is reported to inhibit the activation of HIF-1α or destabilize the dimerization of HIF-1α/β, which leads to the reversal of the MDR effect [74,75,76].

## 4. NO Improves the EPR Effect in Nano-DDS

Maeda and Matsumura reported that proteins and nanoparticles can accumulate in the tumor environment due to the leaky neovascular system and immature lymphatic system—the so-called enhanced permeability and retention (EPR) effect [60,61]. However, several barriers associated with the tumor microenvironment, including abnormal vasculature, high interstitial fluid pressure, tumor-cell density, and abnormal stromal matrix, limit the EPR effect and/or inhibit the permeability of nano-drugs in the cancer tissues [77,78]. Therefore, modulating the blood flow and vasculature at the tumor site is one of the key factors for the improvement of the EPR effect. As described above, the co-delivery of the NO donor with anticancer drugs significantly enhances the efficacy of cancer treatment via the MDR inhibitory effect of NO, including the inhibition of P-gp expression and HIF-1 activation.

Another important characteristic of the NO molecule is “angiogenesis”, which increases the tumoral blood flow and vascular dilation as described above. NO-triggered improvement of the angiogenesis of the tumor environment was studied in order to enhance the EPR effect [79]. Maeda and co-workers reported that several NO generators—such as nitroglycerin, hydroxyurea, and L-Arg—significantly enhanced tumor accumulation of the tested nanomedicines in a mouse model of implanted colon cancer and sarcoma [80]. It is interesting to note that there was no increase in nanoparticle accumulation in the other normal tissues. This is probably due to the difference in vascular structures between intra-tumoral and normal tissue regions. Weidensteiner et al. applied the NO donor JS-K [O(2)-(2,4-dinitrophenyl) 1-[(4-ethoxycarbonyl)piperazin-1-yl]diazen-1-ium-1,2-diolate] to a cerebral blood vessel. They found a significant enhancement in the permeability of the blood–brain tumor barrier in glioma after administration of JS-K, which was confirmed by the magnetic resonance imaging measurement [81]. Tumoral neovascularization was also investigated in the NO donor encapsulated nanocarrier systems. Tahara et al. reported that the tumor accumulation of NO donor-containing PEGylated liposome is twice that of the empty PEGylated liposome, although they show similar blood retention, suggesting the vasodilation effect of NO activity [82]. Several polymeric NO donors have also been designed. A combination of these polymeric NO donors with conventional chemotherapies significantly improved the effectiveness in the treatment of solid tumors [83]. Yoshida et al. reported that co-administration of NO donor and conventional anti-cancer drug significantly improves the therapeutic outcome in clinical trials [84,85]. In these trials, nitroglycerin was used in combination with vinorelbine or cisplatin in the treatment of patients with stage IIIB/IV non-small-cell lung cancer (NSCLC), and the overall response and time to disease progression were significantly improved compared to patients treated with only vinorelbine or cisplatin. Moreover, no remarkable side effects were observed in treated patients. However, a phase III randomized trial of nitroglycerin combination with different chemotherapy on NSCLC in Australia did not give a beneficial outcome [86,87]. This might be due to the different administration timing of nitroglycerin and chemotherapy as well as the heterogeneity of tumor cell lines. More recently, Hu et al. designed an engineered NO-releasing nanocarrier that can respond to and modulate the tumor microenvironment, resulting in a significant enhancement of nanoparticle penetration and heterogeneity distribution in tumor tissue [88]. This multifunctional nanoparticle was composed of a shrinkable hyaluronic acid shell (which is responsive to the highly expressed hyaluronidase in the tumor) modified with an NO donor, a prodrug of doxorubicin and indocyanine green (ICG) as the photothermal agent. ICG generates heat in response to the NIR light irradiation and increases the environmental temperature, which triggers NO release in addition to the hyperthermal effect. Deepagan et al. formulated the NO-generating nanoparticles (NO-NPs) to support the concept of the EPR enhancer [89]. The results in this study clearly showed an increase in the in vivo vasodilation effect and accumulation of co-delivered doxorubicin in the tumor after systemic administration of NO-NPs. Collectively, various nanoplatforms for controlled release of NO have been developed and have shown an effective antitumor therapy through the enhancement of the EPR effect. A big question when using NO-releasing nanomedicine as the EPR enhancer is that whether it can promote tumor growth since NO can enhance the angiogenesis and vascular permeability to supply oxygen and nutrients to the tumor [90,91]. The Maeda group reported that the administration of an NO donor nitroglycerin did not increase the size of the tumor, but it could suppress about 20–30% of tumor growth [80,92]. They explained that the tumor is adapted hypoxia condition, and the energy metabolism is glycolysis-dependent, but not TCA/oxidative phosphorylation-dependent. Besides, permeability enhancement leads the immunological cells more accessible to the tumor tissue. Another important issue is to understand the timing to apply the EPR enhancers during cancer treatment. In the case of NO-releasing nanoparticles, it is known that the accumulation of nanoparticles in tumor tissue by the EPR effect can take place from a few hours to a few days after the systemic administration. Given the unsatisfactory clinical outcomes, however, several controversial and different opinions toward the EPR effect have been reported [93,94]. It can be explained by the poor design and characterization of nanomedicine, and inadequate understanding of tumor pathophysiological progression and the surrounding microenvironment.

## 5. Improvement in NO Bioavailability

Although versatile nanoparticles have been developed to deliver NO to the targeted site and improve the therapeutic effect, spatiotemporal control for the delivery of adequate levels of NO is yet to be achieved. Because NO is a gaseous, free-radical molecule with a very short half-life, once it is endogenously formed or released from nanocarriers, it rapidly disperses and can be metabolized [95,96]. Moreover, the diffusion radius of the NO molecule in biological tissue is reported to be within 100 µm [97]. Another important issue in maintaining the NO level at the target site is avoiding the reaction of NO with ROS [98]. It is reported that NO rapidly reacts with superoxide to form peroxynitrite (ONOO^•−^), which no longer has NO characteristics and induces nitrosative stress [99,100]. Because an excessive amount of ROS is generated in pathological environments, particularly in cancer tissues [101,102], it is worth eliminating the generation of peroxynitrite to maintain an adequate level of NO and inducing NO effects, such as suppression of MDR and improvement of the EPR effect. Several strategies have been attempted to improve the NO bioavailability by increasing endogenous NO synthesis [103], inhibiting the NO metabolism [104], and activating NOS systems [105]. Combination treatment with antioxidants, such as vitamin C and glutathione, has been shown to increase the bioavailability of NO by suppression of oxidative stress and ROS [106,107,108]. We previously confirmed that a combination treatment consisting of an ROS-scavenging polymer with an NO-releasing polymer remarkably improves NO generation and angiogenesis therapeutics [109,110]. We have designed a polycation-PEG-polycation triblock copolymer possessing antioxidant 2,2,6,6-tetramethylpiperidine-1-oxyl (TEMPO) by covalent linkage, which forms a flower-type micelle with conventional polyanions, such as poly(acrylic acid) (PAAc). An NO-releasing triblock copolymer was also designed by PArg-PEG-PArg coupled with PAAc. These two flower-type micelles convert to hydrogel in response to the physiological temperature and ionic strength. When these two micelles are administered together at the inflammation site, the activated macrophage begins to digest the PArg polymer to release NO, while the TEMPO-polymer effectively scavenges ROS. These effects maintain sufficient levels of NO molecules at the site of inflammation and simultaneously suppress the potential for toxic ROS and peroxynitrite formation. Therefore, it is important to consider not only the generation of the NO molecules but also the prevention of NO elimination in order to improve the bioavailability of intact NO molecules during nanoparticle delivery and thus achieve the expected therapeutic effects.

## 6. Conclusions

Targeted NO delivery has recently attracted much attention in the field of cancer therapy because NO is multifunctional and acts in both normal physiological conditions and cancer processes. In this review, we have explained the nanoengineering of NO gas to control tumoral angiogenesis, MDR, HIF-1 expression, the EPR effect, and apoptosis. Although NO is a volatile and gaseous molecule with a very short half-life, it is worth utilizing it for versatile applications. To utilize the highly reactive NO molecule, it is important to spatiotemporally control its bioavailability. Although many publications are documenting the nanoparticle systems for delivery of NO gas and its donors, their efficiency is not optimal, which is mainly due to the gaseous character of the NO molecule in addition to its high reactivity. More sophisticated engineering technology must be developed to successfully deliver NO to the targeted tumor tissue and improve the efficacy of cancer therapeutics. It should be noted that the various effects of NO on the tumor are dependent on the NO concentration in a specific tissue. This the double-edged sword character of NO molecule should be carefully considered for an effective NO-based anticancer therapy. It is also important to take spatiotemporal control of NO level for versatile medical therapeutic applications. Taken together, NO delivery nano-carriers have shown great potential in clinical applications of cancer therapy, although further research is necessary to optimize the delivery strategy and design.

## Figures and Tables

**Figure 1 antioxidants-09-00791-f001:**
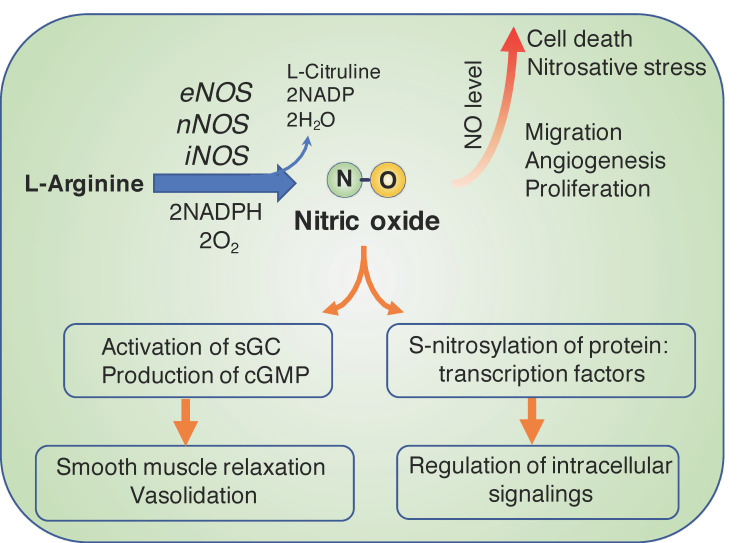
Biosynthesis of nitric oxide (NO) molecule and its biofunctional activities. In the biological system, NO is synthesized from the oxidation of L-arginine (L-Arg) by three nitric oxide synthases (NOSs), namely endothelial (eNOS), neuronal (nNOS), and inducible (iNOS) isoforms. Under physiological conditions, NO activates soluble guanylate cyclase (sGC) to produce cyclic guanidine monophosphate (cGMP), which plays critical roles in relaxation and vasodilation of smooth muscle cells. Additionally, NO causes the S-nitrosylation of proteins, such as transcription factors (such as NF-κB, HIF-1α, EGR-1, etc.), which leads to the downregulation or upregulation of important intracellular signaling pathways [22,23].

**Figure 2 antioxidants-09-00791-f002:**
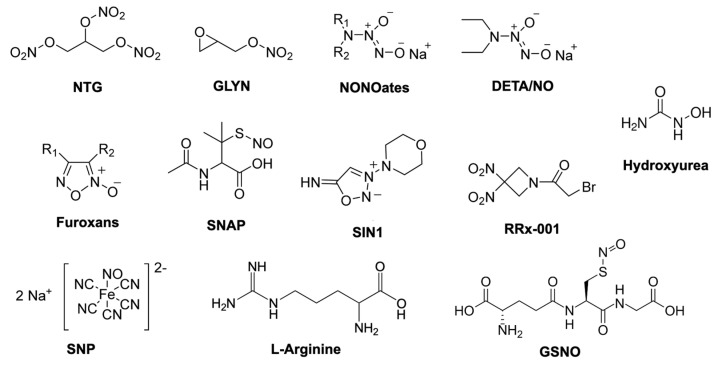
The structure of some important NO donors used in research and clinical settings. NTG (nitroglycerin), GLYN (glycidyl nitrate), NONOates and DETA/NO (*N*-Diazeniumdiolates), SNAP (S-nitroso-N-acetylpenicillamine), SIN1 (3-morpholinosydnonimine), RRx-001 (1-Bromoacetyl-3,3-dinitroazetidine), SNP (sodium nitroprusside), GSNO (S-nitrosoglutathione). Reproduced and modified with permission [25].

**Figure 3 antioxidants-09-00791-f003:**
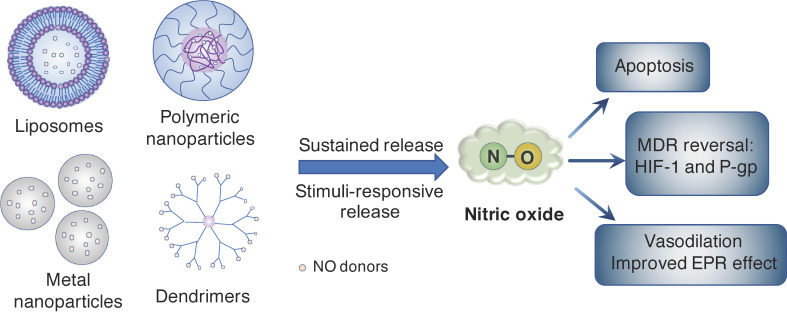
Nanocarriers used to deliver nitric oxide (NO) and the therapeutic effects of NO in cancer. Nanocarriers, such as liposomes, polymeric nanoparticles, metal nanoparticles, and dendrimers, are widely used to improve the stability of NO donors to deliver these molecules to the tumor tissue, where the NO release is triggered by the internal or external stimuli. The NO released from nanocarriers can induce cancer cell apoptosis and death via nitrosative stress, improve the likelihood of reversal of multidrug resistance (MDR) via inhibition of hypoxia-inducible factor-1 (HIF-1) and P-glycoprotein (P-gp) expression, and improve the enhanced permeability and retention (EPR) effect via NO-induced vasodilation.

**Figure 4 antioxidants-09-00791-f004:**
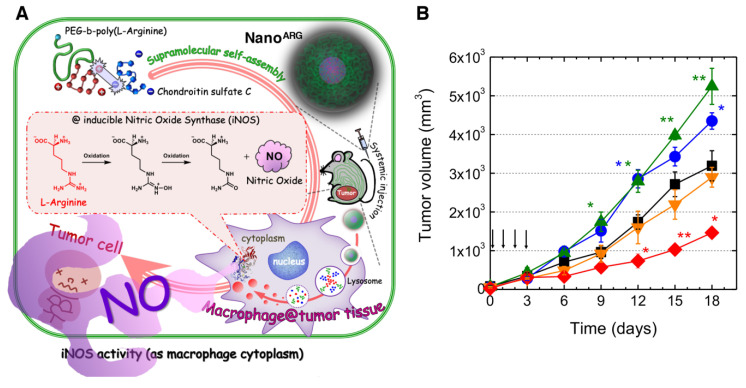
Development of nitric oxide-based anticancer therapeutics using the macrophage-targeted poly(L-arginine) self-assembled nanoparticles (Nano^ARG^). (**A**) The strategic scheme demonstrates that the systemic administration of Nano^ARG^ by activated macrophages leads to the generation of NO in the tumor. After intravenous injection, Nano^ARG^ accumulates at the tumor site via the EPR effect and is internalized by the macrophages, followed by the degradation to L-Arg monomer, which is converted to NO by inducible nitric oxide synthase (iNOS). (**B**) In vivo anticancer efficacy in tumor-bearing mice treated with a different number of Nano^ARG^ injections at the same dosage of L-Arg base (16 mg/kg) (1 injection: blue circle; 2 injections: green triangle; 3 injections: orange triangle; 4 injections: red diamond; untreated: black square). Reproduced with permission [59].

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
