# Peer review of "Nitric Oxide Nano-Delivery Systems for Cancer Therapeutics: Advances and Challenges"

_antioxidants, 2020, doi:10.3390/antiox9090791_

Round 1
Reviewer 1 Report
Presented manuscript concerns use of NO as a theprapeutic agent for cancer diseases. In the study effect of NO on cancer cells has been presented as well as various drug delivery sytems for efficient NO transport have been discussed. In my opinion manuscript is nice wirtten and present important reaserach issues. Howeve, before manuscirpt might be considered for publication some improvements are required.
Properites of the nanmoaterials facilitaiting their use as DDs supports should be more widely presented,
Authors present a couple of ezamples of NO as a drug in cancer treatment and possible ways of its delivery into cancer cells with limitations of these methods. In my opinion some possible ways of overcoming of these problems shoudl also be presented in the manuscript.
Each of the main secrion of the manuscript shudl be end with a brief summary of the presented information.
In my opionio, Authors shoudl present their our opionion and comments more widely in the manuscript. To make review long-lasting and more interesting for the potential readers, Authors shoul add their own comments and opinions on the presented data.
References list should be carefuly checked to meet all of the journal requirememnts. Further, in the references lst more recent study should also be presented, published in 2018 and novel.
Author Response
Answer Reviewer #1’s comments
Presented manuscript concerns use of NO as a theprapeutic agent for cancer diseases. In the study effect of NO on cancer cells has been presented as well as various drug delivery sytems for efficient NO transport have been discussed. In my opinion manuscript is nice wirtten and present important reaserach issues. Howeve, before manuscirpt might be considered for publication some improvements are required.
Answer: Thank you very much for your courteous review and very kind comments. According to your comments, we revised our manuscript. Here we answer to your comments one by one in this sheet as follows (the red colored words or sentences were revised or newly added in the revised manuscript)
Comment 1: Properites of the nanmoaterials facilitaiting their use as DDs supports should be more widely presented,
Answer: Thank you very much for your comment. The main subject of this review article is nitric oxide engineering for new cancer therapy. This is why we have chosen important examples of NO-donor delivery system in this review. We hope you understand.
Comment 2: Authors present a couple of ezamples of NO as a drug in cancer treatment and possible ways of its delivery into cancer cells with limitations of these methods. In my opinion some possible ways of overcoming of these problems shoudl also be presented in the manuscript.
Answer: Thank you very much for your very important comment. As you understand, solutions to the problems are one of the most important points in scientific research. Each researcher deeply investigates and understands the problems and try to find solutions by one's own ideas. This is the reason that we pointed out the problems of the works described in this review article and push readers to come up with new ideas. We hope you understand.
Comment 3: Each of the main secrion of the manuscript shudl be end with a brief summary of the presented information.
Answer: Yes, it is important to summarize the contents in each section. We have briefly described in the text.
Comment 4: In my opionio, Authors should present their our opionion and comments more widely in the manuscript. To make review long-lasting and more interesting for the potential readers, Authors shoul add their own comments and opinions on the presented data.
Answer: Thank you very much for your comment. For the limited spaces, it is not easy to widen the scopes and/or fields in one mini-review. This is the reason why we have limited very important works in this review. As we described in comment 2, it is important to comment on the cited works by our own opinions. We described them in the text. We hope you understand our policy.
Comment 5: References list should be carefuly checked to meet all of the journal requirememnts. Further, in the references lst more recent study should also be presented, published in 2018 and novel.
Answer: Thank you for your comment. We have used Mendeley and carefully checked the format of our references to meet the journal requirements. The dated references are used for foundational information and citation. We also added more contemporary references in the revised manuscript as follows:
- Silkstone, R.S.; Mason, M.G.; Nicholls, P.; Cooper, C.E. Nitrogen dioxide oxidizes mitochondrial cytochrome c. Free Radic. Biol. Med. 2012, 52, 80–87, doi:10.1016/j.freeradbiomed.2011.09.024.
- Heinrich, T.A.; da Silva, R.S.; Miranda, K.M.; Switzer, C.H.; Wink, D.A.; Fukuto, J.M. Biological nitric oxide signalling: chemistry and terminology. Br. J. Pharmacol. 2013, 169, 1417–1429, doi:10.1111/bph.12217.
- Kamm, A.; Przychodzen, P.; Kuban-Jankowska, A.; Jacewicz, D.; Dabrowska, A.M.; Nussberger, S.; Wozniak, M.; Gorska-Ponikowska, M. Nitric oxide and its derivatives in the cancer battlefield. Nitric Oxide 2019, 93, 102–114, doi:https://doi.org/10.1016/j.niox.2019.09.005.
- 2 Huang, Z.; Fu, J.; Zhang, Y. Nitric Oxide Donor-Based Cancer Therapy: Advances and Prospects. J. Med. Chem. 2017, 60, 7617–7635, doi:10.1021/acs.jmedchem.6b01672.
- Mijatović, S.; Savić-Radojević, A.; Plješa-Ercegovac, M.; Simić, T.; Nicoletti, F.; Maksimović-Ivanić, D. The Double-Faced Role of Nitric Oxide and Reactive Oxygen Species in Solid Tumors. Antioxidants (Basel, Switzerland) 2020, 9, 374, doi:10.3390/antiox9050374.
- Ciccarese, F.; Raimondi, V.; Sharova, E.; Silic-Benussi, M.; Ciminale, V. Nanoparticles as Tools to Target Redox Homeostasis in Cancer Cells. Antioxidants (Basel, Switzerland) 2020, 9, 211, doi:10.3390/antiox9030211.
- Cao, Y.; Liu, M.; Cheng, J.; Yin, J.; Huang, C.; Cui, H.; Zhang, X.; Zhao, G. Acidity-Triggered Tumor-Targeted Nanosystem for Synergistic Therapy via a Cascade of ROS Generation and NO Release. ACS Appl. Mater. Interfaces 2020, 12, 28975–28984, doi:10.1021/acsami.0c04791.
- Jiang, D.; Yue, T.; Wang, G.; Wang, C.; Chen, C.; Cao, H.; Gao, Y. Peroxynitrite (ONOO−) generation from the HA-TPP@NORM nanoparticles based on synergistic interactions between nitric oxide and photodynamic therapies for elevating anticancer efficiency. New J. Chem. 2020, 44, 162–170, doi:10.1039/C9NJ04763H.
- Kim, D.E.; Kim, C.W.; Lee, H.J.; Min, K.H.; Kwack, K.H.; Lee, H.-W.; Bang, J.; Chang, K.; Lee, S.C. Intracellular NO-Releasing Hyaluronic Acid-Based Nanocarriers: A Potential Chemosensitizing Agent for Cancer Chemotherapy. ACS Appl. Mater. Interfaces 2018, 10, 26870–26881, doi:10.1021/acsami.8b06848.
- Sun, F.; Wang, Y.; Luo, X.; Ma, Z.; Xu, Y.; Zhang, X.; Lv, T.; Zhang, Y.; Wang, M.; Huang, Z.; et al. Anti-CD24 Antibody–Nitric Oxide Conjugate Selectively and Potently Suppresses Hepatic Carcinoma. Cancer Res. 2019, 79, 3395 LP – 3405, doi:10.1158/0008-5472.CAN-18-2839.
- Davidson, A.; Veillard, A.-S.; Tognela, A.; Chan, M.M.K.; Hughes, B.G.M.; Boyer, M.; Briscoe, K.; Begbie, S.; Abdi, E.; Crombie, C.; et al. A phase III randomized trial of adding topical nitroglycerin to first-line chemotherapy for advanced nonsmall-cell lung cancer: the Australasian lung cancer trials group NITRO trial. Ann. Oncol. 2015, 26, 2280–2286, doi:https://doi.org/10.1093/annonc/mdv373.
- de Jong, E.E.C.; van Elmpt, W.; Leijenaar, R.T.H.; Hoekstra, O.S.; Groen, H.J.M.; Smit, E.F.; Boellaard, R.; van der Noort, V.; Troost, E.G.C.; Lambin, P.; et al. [18F]FDG PET/CT-based response assessment of stage IV non-small cell lung cancer treated with paclitaxel-carboplatin-bevacizumab with or without nitroglycerin patches. Eur. J. Nucl. Med. Mol. Imaging 2017, 44, 8–16, doi:10.1007/s00259-016-3498-y
- Deepagan, V.G.; Ko, H.; Kwon, S.; Rao, N.V.; Kim, S.K.; Um, W.; Lee, S.; Min, J.; Lee, J.; Choi, K.Y.; et al. Intracellularly Activatable Nanovasodilators To Enhance Passive Cancer Targeting Regime. Nano Lett. 2018, 18, 2637–2644, doi:10.1021/acs.nanolett.8b00495.
- Maeda, H.; Tsukigawa, K.; Fang, J. A Retrospective 30 Years After Discovery of the Enhanced Permeability and Retention Effect of Solid Tumors: Next-Generation Chemotherapeutics and Photodynamic Therapy—Problems, Solutions, and Prospects. Microcirculation 2016, 23, 173–182, doi:10.1111/micc.12228.
- Fang, J.; Islam, W.; Maeda, H. Exploiting the dynamics of the EPR effect and strategies to improve the therapeutic effects of nanomedicines by using EPR effect enhancers. Adv. Drug Deliv. Rev. 2020, doi:https://doi.org/10.1016/j.addr.2020.06.005.
- Seki, T.; Fang, J.; Maeda, H. Enhanced delivery of macromolecular antitumor drugs to tumors by nitroglycerin application. Cancer Sci. 2009, 100, 2426–2430, doi:10.1111/j.1349-7006.2009.01323.x.
- Kwon, I.K.; Lee, S.C.; Han, B.; Park, K. Analysis on the current status of targeted drug delivery to tumors. J. Control. Release 2012, 164, 108–114, doi:https://doi.org/10.1016/j.jconrel.2012.07.010.
- Nichols, J.W.; Bae, Y.H. EPR: Evidence and fallacy. J. Control. Release 2014, 190, 451–464, doi:https://doi.org/10.1016/j.jconrel.2014.03.057.
We are happy if you understand our explanation and accept our revised manuscript for publication.
Reviewer 2 Report
In this manuscript by Vong and Nagasaki, the authors provide a brief review of nitric oxide (NO) in cancer and NO-delivering nanocarriers as potential cancer therapeutics. Such is an essential field of investigation, as evident by the number of papers recently published on the subject (547 since 2018, PubMed search “nitric oxide” + “cancer” + “therapeutics”). Generally, their manuscript is well written, concise, yet provides sufficient background to understand the underlying problems current information on NO-generating nanoparticles, and additional insights into some lesser-known aspects of NO-affected cancer therapy (e.g., combating multidrug resistance, enhanced permeability, and retention). However, I had a few concerns that, if addressed, could improve the significance and impact of this paper once published.
[Line 60] A major concern I have with this review is its generally dated references. Of its 87 references, only 16 are from 2018 or later. For example, the references for recent cancer treatments using NO are reviews from 2010 and 2016. I can think of recent reviews, e.g., Kamm et al., 2019, Nitric Oxide, that would be far more appropriate. This is something that should be examined throughout their manuscript.
[Figure 1] It may merely be a problem with my pdf, but some text (L-arginine, eNOS nNOS NOS, Nitric oxide) are not rendered correctly. You may wish to check this throughout.
[Line 72] Which transcription factors? p53?
[Line 136] NIF activated NO-releasing nanoparticles is a more recent advance in the field. Thank you for highlighting it.
[Line 163] While reporting the Fan et al. work with organosilica nanoparticles containing glucose oxidase and L-arginine is worthwhile, I have always been perplexed by their rationale. High glucose concentrations are thought to exist within many cancer cells as part of the Warburg effect, but not in their immediate environment. Likewise, isn’t the usual cause for peroxide in the tumor microenvironment resulting from infiltrating macrophages, not the cancer cells themselves. You may wish to comment on these.
[Line 175] As anticipated, work from the author’s laboratory is showcased at this point. Reasonable and appropriate.
[Line 194] Perhaps one of the most important take-home lessons in this review: low local levels of NO can promote tumor growth, ostensibly through angiogenesis, whereas high levels of NO can be cytotoxic. This yin-yang aspect of NO when it comes to cancer therapeutics needs to be emphasized, as it is different than its role in other treatments (e.g., cardiovascular disease).
[Figure 3A] This figure was not rendered correctly. Please redo.
[Line 216] This and the following section are most important, insofar as they point out that even lower NO concentrations delivered locally can have a positive therapeutic effect on cancer through phenomena such as P-glycoprotein suppression and increased vascular permeability. In both cases, these need to be weighed against tumor-beneficial phenomena promoted by low NO concentrations (e.g., angiogenesis promotion). More examples need to be provided to show which dominates under real-world conditions.
[Line 284] I am confused regarding the need for ROS suppression. ROS promotes peroxynitrite formation, which is important for the cytotoxic effects of NO. Are you trying to say that the impacts of low NO concentrations (MDR suppression, increased EPR) are more important than its cytotoxicity, especially given the counterbalancing effects of angiogenesis promotion? There needs to be more clarity here.
[References] Again, as stated above, many of the references are rather dated. Unless they are seminal, foundational papers, your review would benefit from more contemporary references.
Author Response
Answer Reviewer #2’s comments
In this manuscript by Vong and Nagasaki, the authors provide a brief review of nitric oxide (NO) in cancer and NO-delivering nanocarriers as potential cancer therapeutics. Such is an essential field of investigation, as evident by the number of papers recently published on the subject (547 since 2018, PubMed search “nitric oxide” + “cancer” + “therapeutics”). Generally, their manuscript is well written, concise, yet provides sufficient background to understand the underlying problems current information on NO-generating nanoparticles, and additional insights into some lesser-known aspects of NO-affected cancer therapy (e.g., combating multidrug resistance, enhanced permeability, and retention). However, I had a few concerns that, if addressed, could improve the significance and impact of this paper once published.
Answer: Thank you very much for your courteous review and very kind comments. According to your comments, we revised our manuscript. Here we answer to your comments one by one in this sheet as follows (the red colored words or sentences were revised or newly added in the revised manuscript)
Comment 1: [Line 60] A major concern I have with this review is its generally dated references. Of its 87 references, only 16 are from 2018 or later. For example, the references for recent cancer treatments using NO are reviews from 2010 and 2016. I can think of recent reviews, e.g., Kamm et al., 2019, Nitric Oxide, that would be far more appropriate. This is something that should be examined throughout their manuscript.
Answer: Thank you for your suggestion. We added the reference as follows:
Original (page 61-62)
Because NO has versatile functions in the cancer microenvironment, versatile and unique cancer treatments using NO have been developed recently.[13][17]
Revised (page 60-64)
In the physiological condition, NO basically undergoes a rapid oxidation to form nitrite anion (NO2–), which is further oxidized to nitrate anion (NO3–). The autooxidation of NO molecules also occurs to generate N2O3 via formation of nitrite radical (NO2•), and this is reversible reaction.[17][18] Because NO has versatile functions in the cancer microenvironment, versatile and unique cancer treatments using NO have been developed recently.[13][19][20]
- Silkstone, R.S.; Mason, M.G.; Nicholls, P.; Cooper, C.E. Nitrogen dioxide oxidizes mitochondrial cytochrome c. Free Radic. Biol. Med. 2012, 52, 80–87, doi:10.1016/j.freeradbiomed.2011.09.024.
- Heinrich, T.A.; da Silva, R.S.; Miranda, K.M.; Switzer, C.H.; Wink, D.A.; Fukuto, J.M. Biological nitric oxide signalling: chemistry and terminology. Br. J. Pharmacol. 2013, 169, 1417–1429, doi:10.1111/bph.12217.
- Kamm, A.; Przychodzen, P.; Kuban-Jankowska, A.; Jacewicz, D.; Dabrowska, A.M.; Nussberger, S.; Wozniak, M.; Gorska-Ponikowska, M. Nitric oxide and its derivatives in the cancer battlefield. Nitric Oxide 2019, 93, 102–114, doi:https://doi.org/10.1016/j.niox.2019.09.005.
Comment 2: [Figure 1] It may merely be a problem with my pdf, but some text (L-arginine, eNOS nNOS NOS, Nitric oxide) are not rendered correctly. You may wish to check this throughout.
Answer: We have checked Figure 1 carefully. Those texts are rendered correctly in Figure and explained in Figure caption as follows:
Figure 1: Biosynthesis of nitric oxide (NO) molecule and its biofunctional activities. In the biological system, NO is synthesized from the oxidation of L-arginine (L-Arg) by three nitric oxide synthases (NOSs), namely endothelial (eNOS), neuronal (nNOS), and inducible (iNOS) isoforms. Under physiological conditions, NO activates soluble guanylate cyclase (sGC) to produce cyclic guanidine monophosphate (cGMP), which plays critical roles in relaxation and vasodilation of smooth muscle cells. Additionally, NO causes the S-nitrosylation of proteins such as transcription factors (such as NF-kB, HIF-1a, EGR-1, etc…), which leads to the downregulation or upregulation of important intracellular signaling pathways.[22][23]
[22] Sha, Y.; Marshall, H.E. S-nitrosylation in the regulation of gene transcription. Biochim. Biophys. Acta - Gen. Subj. 2012, 1820, 701–711, doi:https://doi.org/10.1016/j.bbagen.2011.05.008.
[23] Kumar, S.; Singh, R.K.; Bhardwaj, T.R. Therapeutic role of nitric oxide as emerging molecule. Biomed. Pharmacother. 2017, 85, 182–201, doi:https://doi.org/10.1016/j.biopha.2016.11.125.
Comment 3: [Line 72] Which transcription factors? p53?
Answer: We added some well-known transcription factors and references on the S-nitrosylation of protein by NO. The text was revised as follows:
Original (Line 70-76)
Figure 1: Biosynthesis of nitric oxide (NO) molecule and its biofunctional activities. In the biological system, NO is synthesized from the oxidation of L-arginine (L-Arg) by three nitric oxide synthases (NOSs), namely endothelial (eNOS), neuronal (nNOS), and inducible (iNOS) isoforms. Under physiological conditions, NO activates soluble guanylate cyclase (sGC) to produce cyclic guanidine monophosphate (cGMP), which plays critical roles in relaxation and vasodilation of smooth muscle cells. Additionally, NO causes the S-nitrosylation of proteins such as transcription factors, which leads to the downregulation or upregulation of important intracellular signaling pathways.
Revised (Line 72-78)
Figure 1: Biosynthesis of nitric oxide (NO) molecule and its biofunctional activities. In the biological system, NO is synthesized from the oxidation of L-arginine (L-Arg) by three nitric oxide synthases (NOSs), namely endothelial (eNOS), neuronal (nNOS), and inducible (iNOS) isoforms. Under physiological conditions, NO activates soluble guanylate cyclase (sGC) to produce cyclic guanidine monophosphate (cGMP), which plays critical roles in relaxation and vasodilation of smooth muscle cells. Additionally, NO causes the S-nitrosylation of proteins such as transcription factors (such as NF-kB, HIF-1a, EGR-1, etc…), which leads to the downregulation or upregulation of important intracellular signaling pathways.[22][23]
[22] Sha, Y.; Marshall, H.E. S-nitrosylation in the regulation of gene transcription. Biochim. Biophys. Acta - Gen. Subj. 2012, 1820, 701–711, doi:https://doi.org/10.1016/j.bbagen.2011.05.008.
[23] Kumar, S.; Singh, R.K.; Bhardwaj, T.R. Therapeutic role of nitric oxide as emerging molecule. Biomed. Pharmacother. 2017, 85, 182–201, doi:https://doi.org/10.1016/j.biopha.2016.11.125.
Comment 4. [Line 136] NIF activated NO-releasing nanoparticles is a more recent advance in the field. Thank you for highlighting it.
Answer: Thank you for your comment. These works are important for readers.
Comment 5. [Line 163] While reporting the Fan et al. work with organosilica nanoparticles containing glucose oxidase and L-arginine is worthwhile, I have always been perplexed by their rationale. High glucose concentrations are thought to exist within many cancer cells as part of the Warburg effect, but not in their immediate environment. Likewise, isn’t the usual cause for peroxide in the tumor microenvironment resulting from infiltrating macrophages, not the cancer cells themselves. You may wish to comment on these.
Answer: Thank you for your comment and information. As your suggestion, we added references and revised the manuscript as follows:
Original (Line 166-169)
As mentioned above, however, the oxidation reaction of L-Arg to chemically form NO by H2O2 is not clearly confirmed, and a high concentration of reactants and catalysts is required to achieve rapid and efficient NO generation.[38][39]
Revised (Line 182-190)
As mentioned above, however, the oxidation reaction of L-Arg to chemically form NO by H2O2 is not clearly confirmed, and a high concentration of reactants and catalysts is required to achieve rapid and efficient NO generation.[50][51] Besides, an increased uptake of glucose by Warburg effect enhances the intracellular glucose concentration in cancer cells, but not surrounding tumor microenvironment;[52][53] thus, it should be considered to deliver sufficiently both L-Arg substrate and GOX enzyme into the cancer cells. However, the other mechanisms might not be considered that the delivered L-Arg can be catalyzed into NO in the tumor microenvironment by the overexpressed iNOS from tumor-associated macrophage.
[52] Sun, H.; Chen, L.; Cao, S.; Liang, Y.; Xu, Y. Warburg Effects in Cancer and Normal Proliferating Cells: Two Tales of the Same Name. Genomics. Proteomics Bioinformatics 2019, 17, 273–286, doi:https://doi.org/10.1016/j.gpb.2018.12.006.
[53]. Fadaka, A.; Ajiboye, B.; Ojo, O.; Adewale, O.; Olayide, I.; Emuowhochere, R. Biology of glucose metabolization in cancer cells. J. Oncol. Sci. 2017, 3, 45–51, doi:https://doi.org/10.1016/j.jons.2017.06.002.
Comment 6. [Line 175] As anticipated, work from the author’s laboratory is showcased at this point. Reasonable and appropriate.
Answer: Thank you very much for your comment and understanding. We are very much happy with your comment like this.
Comment 7. [Line 194] Perhaps one of the most important take-home lessons in this review: low local levels of NO can promote tumor growth ostensibly through angiogenesis whereas high tumor growth, ostensibly through angiogenesis, whereas high levels of NO can be cytotoxic. This yin-yang aspect of NO when it comes to cancer therapeutics needs to be emphasized, as it is different than its role in other treatments (e.g., cardiovascular disease).
Answer: We are happy that you understand the important lesson that we proposed in this review paper. We have thoroughly emphasized the double-faced role of NO in cancer therapy in our revised manuscript.
Section 1. Line 52-61
Section 2: Line 198-206
Section 4: Line 255-256
Section 5: Line 317-319
Section 6: Line 361-364 was newly added as follows:
It should be noted that the various effects of NO on the tumor are dependent on the NO concentration in a specific tissue. This double-edged sword character of NO molecule should be carefully considered for an effective NO-based anticancer therapy. It is also important to spatiotemporal control of NO level for versatile medical therapeutic applications.
Comment 8. [Figure 3A] This figure was not rendered correctly. Please redo.
Answer: Thank you for your comment. We are sorry we could not render it well. The figure has been modified as follows:
Figure 4: Development of nitric oxide-based anticancer therapeutics using the macrophage-targeted poly(L-arginine) self-assembled nanoparticles (NanoARG). (A) The strategic scheme demonstrates that the systemic administration of NanoARG by activated macrophages leads to the generation of NO in the tumor. After intravenous injection, NanoARG accumulates at the tumor site via the EPR effect and is internalized by the macrophages, followed by the degradation to L-Arg monomer, which is converted to NO by inducible nitric oxide synthase (iNOS). (B) In vivo anticancer efficacy in tumor-bearing mice treated with a different number of NanoARG injections at same dosage of L-Arg base (16 mg/kg) (1 injection: blue circle; 2 injections: green triangle; 3 injections: orange triangle; 4 injections: red diamond; untreated: black square). Reproduced with permission [59].
Comment 9. [Line 216] This and the following section are most important, insofar as they point out that even lower NO concentrations delivered locally can have a positive therapeutic effect on cancer through phenomena such as P-glycoprotein suppression and increased vascular permeability. In both cases, these need to be weighed against tumor-beneficial phenomena promoted by low NO concentrations (e.g., angiogenesis promotion). More examples need to be provided to show which dominates under real-world conditions.
Answer: Thank you for your comment. To clarify this point, we added the discussion in the revised manuscript as follows:
Original (Line 280-283)
ICG generates heat in response to the NIR light irradiation and increases the environmental temperature, which triggers NO release in addition to the hyperthermal effect. Collectively, various nanoplatforms for controlled release of NO have been developed and have shown an effective antitumor therapy through the enhancement of the EPR effect.
Revised (Line 299-319)
ICG generates heat in response to the NIR light irradiation and increases the environmental temperature, which triggers NO release in addition to the hyperthermal effect. Deepagan, et al. formulated the NO-generating nanoparticles (NO-NPs) to support the concept of the EPR enhancer.[89] The results in this study clearly showed an increase in the in vivo vasolidation effect and accumulation of co-delivered doxorubicin in the tumor after systemic administration of NO-NPs. Collectively, various nanoplatforms for controlled release of NO have been developed and have shown an effective antitumor therapy through the enhancement of the EPR effect. A big question when using NO-releasing nanomedicine as EPR enhancer is that whether it can promote tumor growth since NO can enhance the angiogenesis and vascular permeability to supply oxygen and nutrient to tumor.[90][91] However, Maeda group reported that administration of an NO dornor nitroglycerin did not increase the size of tumor, but it could suppress about 20-30% of tumor growth.[80][92] They explained that tumor is adapted hypoxia condition and the energy metabolism is glycolysis dependent, but not TCA/oxidative phosphorylation dependent. Additionally, permeability enhancement leads the immunological cells more accessible to the tumor tissue. Another improtant issue is to understand the timing to apply the EPR enhancers during cancer treatment. In case of NO-releasing nanoparticles, it is known that accumulation of nanoparticle in tumor tissue by EPR effect can be taken place from few hours to few days after sysmetic adminitration. However, due to the unstatisfactory clinical outcomes, some controversial different opinions toward EPR effect has been reported.[93][94] It can be explained by the poor design and charaterization of nanomedicine, and inadequate understanding of tumor pathophysiological progression and surrounding microenvironment.
- Deepagan, V.G.; Ko, H.; Kwon, S.; Rao, N.V.; Kim, S.K.; Um, W.; Lee, S.; Min, J.; Lee, J.; Choi, K.Y.; et al. Intracellularly Activatable Nanovasodilators To Enhance Passive Cancer Targeting Regime. Nano Lett. 2018, 18, 2637–2644, doi:10.1021/acs.nanolett.8b00495.
- Maeda, H.; Tsukigawa, K.; Fang, J. A Retrospective 30 Years After Discovery of the Enhanced Permeability and Retention Effect of Solid Tumors: Next-Generation Chemotherapeutics and Photodynamic Therapy—Problems, Solutions, and Prospects. Microcirculation 2016, 23, 173–182, doi:10.1111/micc.12228.
- Fang, J.; Islam, W.; Maeda, H. Exploiting the dynamics of the EPR effect and strategies to improve the therapeutic effects of nanomedicines by using EPR effect enhancers. Adv. Drug Deliv. Rev. 2020, doi:https://doi.org/10.1016/j.addr.2020.06.005.
- Seki, T.; Fang, J.; Maeda, H. Enhanced delivery of macromolecular antitumor drugs to tumors by nitroglycerin application. Cancer Sci. 2009, 100, 2426–2430, doi:10.1111/j.1349-7006.2009.01323.x.
- Kwon, I.K.; Lee, S.C.; Han, B.; Park, K. Analysis on the current status of targeted drug delivery to tumors. J. Control. Release 2012, 164, 108–114, doi:https://doi.org/10.1016/j.jconrel.2012.07.010.
- Nichols, J.W.; Bae, Y.H. EPR: Evidence and fallacy. J. Control. Release 2014, 190, 451–464, doi:https://doi.org/10.1016/j.jconrel.2014.03.057.
Comment 10. [Line 284] I am confused regarding the need for ROS suppression. ROS promotes peroxynitrite formation, which is important for the cytotoxic effects of NO. Are you trying to say that the impacts of low NO concentrations (MDR suppression, increased EPR) are more important than its cytotoxicity, especially given the counterbalancing effects of angiogenesis promotion? There needs to be more clarity here.
Answer: We are sorry for confusing our manuscript at this point. We wanted to increase the effect of NO, suppressing the toxicity of ROS and possible peroxynitrite for the angiogenesis treatment. To clarify this point, we added the references and discussion in the revised manuscript as follows:
Original (Line 306-312)
These two flower-type micelles convert to hydrogel in response to the physiological temperature and ionic strength. When these two micelles are administered together at the inflammation site, the activated macrophage begins to digest the PArg gel to release NO, while the TEMPO-gel effectively scavenges ROS. Therefore, it is important to consider not only the generation of the NO molecules but also the prevention of NO elimination in order to improve the bioavailability of intact NO molecules during nanoparticle delivery and thus achieve the expected therapeutic effects.
Revised (Line 342-349)
These two flower-type micelles convert to hydrogel in response to the physiological temperature and ionic strength. When these two micelles are administered together at the inflammation site, the activated macrophage begins to digest the PArg polymer to release NO, while the TEMPO-polymer effectively scavenges ROS. These effects maintain sufficient levels of NO molecules at the site of inflammation and simultaneously suppress the potential for toxic ROS and peroxynitrite formation. Therefore, it is important to consider not only the generation of the NO molecules but also the prevention of NO elimination in order to improve the bioavailability of intact NO molecules during nanoparticle delivery and thus achieve the expected therapeutic effects.
Comment 11. [References] Again, as stated above, many of the references are rather dated. Unless they are seminal, foundational papers, your review would benefit from more contemporary references.
Answer: Thank you for your comment. We are using the dated references for the foundational information and citation. We also added more contemporary references in the revised manuscript as follows:
- Silkstone, R.S.; Mason, M.G.; Nicholls, P.; Cooper, C.E. Nitrogen dioxide oxidizes mitochondrial cytochrome c. Free Radic. Biol. Med. 2012, 52, 80–87, doi:10.1016/j.freeradbiomed.2011.09.024.
- Heinrich, T.A.; da Silva, R.S.; Miranda, K.M.; Switzer, C.H.; Wink, D.A.; Fukuto, J.M. Biological nitric oxide signalling: chemistry and terminology. Br. J. Pharmacol. 2013, 169, 1417–1429, doi:10.1111/bph.12217.
- Kamm, A.; Przychodzen, P.; Kuban-Jankowska, A.; Jacewicz, D.; Dabrowska, A.M.; Nussberger, S.; Wozniak, M.; Gorska-Ponikowska, M. Nitric oxide and its derivatives in the cancer battlefield. Nitric Oxide 2019, 93, 102–114, doi:https://doi.org/10.1016/j.niox.2019.09.005.
- 2 Huang, Z.; Fu, J.; Zhang, Y. Nitric Oxide Donor-Based Cancer Therapy: Advances and Prospects. J. Med. Chem. 2017, 60, 7617–7635, doi:10.1021/acs.jmedchem.6b01672.
- Mijatović, S.; Savić-Radojević, A.; Plješa-Ercegovac, M.; Simić, T.; Nicoletti, F.; Maksimović-Ivanić, D. The Double-Faced Role of Nitric Oxide and Reactive Oxygen Species in Solid Tumors. Antioxidants (Basel, Switzerland) 2020, 9, 374, doi:10.3390/antiox9050374.
- Ciccarese, F.; Raimondi, V.; Sharova, E.; Silic-Benussi, M.; Ciminale, V. Nanoparticles as Tools to Target Redox Homeostasis in Cancer Cells. Antioxidants (Basel, Switzerland) 2020, 9, 211, doi:10.3390/antiox9030211.
- Cao, Y.; Liu, M.; Cheng, J.; Yin, J.; Huang, C.; Cui, H.; Zhang, X.; Zhao, G. Acidity-Triggered Tumor-Targeted Nanosystem for Synergistic Therapy via a Cascade of ROS Generation and NO Release. ACS Appl. Mater. Interfaces 2020, 12, 28975–28984, doi:10.1021/acsami.0c04791.
- Jiang, D.; Yue, T.; Wang, G.; Wang, C.; Chen, C.; Cao, H.; Gao, Y. Peroxynitrite (ONOO−) generation from the HA-TPP@NORM nanoparticles based on synergistic interactions between nitric oxide and photodynamic therapies for elevating anticancer efficiency. New J. Chem. 2020, 44, 162–170, doi:10.1039/C9NJ04763H.
- Kim, D.E.; Kim, C.W.; Lee, H.J.; Min, K.H.; Kwack, K.H.; Lee, H.-W.; Bang, J.; Chang, K.; Lee, S.C. Intracellular NO-Releasing Hyaluronic Acid-Based Nanocarriers: A Potential Chemosensitizing Agent for Cancer Chemotherapy. ACS Appl. Mater. Interfaces 2018, 10, 26870–26881, doi:10.1021/acsami.8b06848.
- Sun, F.; Wang, Y.; Luo, X.; Ma, Z.; Xu, Y.; Zhang, X.; Lv, T.; Zhang, Y.; Wang, M.; Huang, Z.; et al. Anti-CD24 Antibody–Nitric Oxide Conjugate Selectively and Potently Suppresses Hepatic Carcinoma. Cancer Res. 2019, 79, 3395 LP – 3405, doi:10.1158/0008-5472.CAN-18-2839.
- Davidson, A.; Veillard, A.-S.; Tognela, A.; Chan, M.M.K.; Hughes, B.G.M.; Boyer, M.; Briscoe, K.; Begbie, S.; Abdi, E.; Crombie, C.; et al. A phase III randomized trial of adding topical nitroglycerin to first-line chemotherapy for advanced nonsmall-cell lung cancer: the Australasian lung cancer trials group NITRO trial. Ann. Oncol. 2015, 26, 2280–2286, doi:https://doi.org/10.1093/annonc/mdv373.
- de Jong, E.E.C.; van Elmpt, W.; Leijenaar, R.T.H.; Hoekstra, O.S.; Groen, H.J.M.; Smit, E.F.; Boellaard, R.; van der Noort, V.; Troost, E.G.C.; Lambin, P.; et al. [18F]FDG PET/CT-based response assessment of stage IV non-small cell lung cancer treated with paclitaxel-carboplatin-bevacizumab with or without nitroglycerin patches. Eur. J. Nucl. Med. Mol. Imaging 2017, 44, 8–16, doi:10.1007/s00259-016-3498-y
- Deepagan, V.G.; Ko, H.; Kwon, S.; Rao, N.V.; Kim, S.K.; Um, W.; Lee, S.; Min, J.; Lee, J.; Choi, K.Y.; et al. Intracellularly Activatable Nanovasodilators To Enhance Passive Cancer Targeting Regime. Nano Lett. 2018, 18, 2637–2644, doi:10.1021/acs.nanolett.8b00495.
- Maeda, H.; Tsukigawa, K.; Fang, J. A Retrospective 30 Years After Discovery of the Enhanced Permeability and Retention Effect of Solid Tumors: Next-Generation Chemotherapeutics and Photodynamic Therapy—Problems, Solutions, and Prospects. Microcirculation 2016, 23, 173–182, doi:10.1111/micc.12228.
- Fang, J.; Islam, W.; Maeda, H. Exploiting the dynamics of the EPR effect and strategies to improve the therapeutic effects of nanomedicines by using EPR effect enhancers. Adv. Drug Deliv. Rev. 2020, doi:https://doi.org/10.1016/j.addr.2020.06.005.
- Seki, T.; Fang, J.; Maeda, H. Enhanced delivery of macromolecular antitumor drugs to tumors by nitroglycerin application. Cancer Sci. 2009, 100, 2426–2430, doi:10.1111/j.1349-7006.2009.01323.x.
- Kwon, I.K.; Lee, S.C.; Han, B.; Park, K. Analysis on the current status of targeted drug delivery to tumors. J. Control. Release 2012, 164, 108–114, doi:https://doi.org/10.1016/j.jconrel.2012.07.010.
- Nichols, J.W.; Bae, Y.H. EPR: Evidence and fallacy. J. Control. Release 2014, 190, 451–464, doi:https://doi.org/10.1016/j.jconrel.2014.03.057.
We are happy if you understand our explanation and accept our revised manuscript for publication.

Reviewer 3 Report
The proposed review describes the role of nano systems for delivery of NO for cancer treatment. The topic is very interesting and worth writing a review paper, nevertheless several improvements are required before publication on Antioxidants.
1) page 1 chapter 1. The NO biochemistry should be integrated with description of auto-oxidation reactions (such as NO2 and N2O3 production)
2) page 2 to improve clarity in this chapter, a figure with structures of principal classes of NO donors should be added, furthermore at least one reference related to this point (e.g. J. Med. Chem. 2017, 60, 7617−7635) should also be added.
Among the NO donors, please cite the most promising in clinical trials and also add the drawbacks of these approaches (Toxicity, tolerance)
3) page 5 improve the clarity and quality of figure 3, panel A
4) page 6 regarding EPR effect, please cite and carefully evaluate the Maeda’s extensive review Maeda, H., Tsukigawa, K., Fang, J. A Retrospective 30 Years After Discovery of the Enhanced Permeability and Retention Effect of Solid Tumors: Next-Generation Chemotherapeutics and Photodynamic Therapy—Problems, Solutions, and Prospects (2016) Microcirculation, 23 (3), pp. 173-182.
5) page 7 regarding the role of nitroglycerin in cancer treatment: further clinical results on nitroglycerin plus anticancer drugs better describe the real advantages, and limits, of this approach : Annals of Oncology 2015, 26(11), pp. 2280-2286 and European Journal of Nuclear Medicine and Molecular Imaging 2017
44(1), pp. 8-16
6) A table summarizing the preclinical or clinical results should be added, indicating the tumor model, NP system, dosage, NO released concentration and principal results
7) recent works on NP NO delivery and NO application on cancers should be cited and discussed:e.g. ACS Applied Materials and Interfaces 10, 2018, 26870-26881 Intracellular NO-Releasing Hyaluronic Acid-Based Nanocarriers: A Potential Chemosensitizing Agent for Cancer Chemotherapy
European Journal of Pharmacology 826, 2018, 158-168 Nitric oxide donors for prostate and bladder cancers: Current state and challenges
Cancer Research 79, 2019, Pages 3395-3405 Anti-CD24 antibody-nitric oxide conjugate selectively and potently suppresses hepatic carcinoma (this may be of particular interest for evaluating the NO targeting approach)
8) Conclusion chapter. Due to the experience and skill of the authors in NO delivery a more critical evaluation about different approaches, NO donors is awaited. Specifically about the term 'More sophisticated engineering technology' , please take into account, and cite, the problems of scale-up production, GMP and regulatory requirements for a clinical grade medicine.
Author Response

(The authors gave the same response as above.)

Round 2
Reviewer 1 Report
I have carefully checked the revised manuscript. Authors have properly adressed my comments and I agree with Authors explanations. The information presented in the manuscript has been updated as weel as cited references have been expanded. In my opinion manuscript might be accepted for publication in the present form.
Reviewer 3 Report
The revised version of the manuscript has addressed the previous critics and is now acceptable for publication.